# Coenzyme Q10 and Xenobiotic Metabolism: An Overview

**DOI:** 10.3390/ijms26125788

**Published:** 2025-06-17

**Authors:** David Mantle, Beatrice A. Golomb

**Affiliations:** 1Pharma Nord (UK) Ltd., Morpeth NE61 2DB, Northumberland, UK; 2San Diego School of Medicine, University of California, La Jolla, CA 92093, USA; bgolomb@health.ucsd.edu

**Keywords:** coenzyme Q10, xenobiotics, pesticides, heavy metals, industrial solvents, endocrine disrupters, carcinogens, pharmacological drugs, lifestyle toxins

## Abstract

Mitochondria are primary targets for environmental toxic chemicals; these typically disrupt the mitochondrial electron transport chain, resulting in reduced ATP production, increased reactive oxygen free radical species (ROS)-induced oxidative stress, increased apoptosis, and increased inflammation. This in turn suggests a rationale for investigating the potential role of coenzyme Q10 (CoQ10) in mediating such chemical-induced mitochondrial dysfunction, given the key roles of CoQ10 in promoting normal mitochondrial function, and as an antioxidant and anti-apoptotic and anti-inflammatory agent. In the present article, we have, therefore, reviewed the potential role of supplementary CoQ10 in improving mitochondrial function and mediating adverse effects following exposure to a number of environmental toxins, including pesticides, heavy metals, industrial solvents, endocrine-disrupting agents, and carcinogens, as well as pharmacological drugs and lifestyle toxicants.

## 1. Introduction

The human body can be impacted by exposure to a wide range of chemical substances, including pesticides, heavy metals, industrial solvents, endocrine-disrupting agents, and carcinogens, as well as pharmacological drugs and lifestyle toxicants. At the cellular level, most of these chemicals produce toxicity in part through mitochondrial impairment and oxidative stress. Mitochondria have a key role in normal cell metabolism, including the production of ATP, the generation of oxidising free radical species (ROS), calcium homeostasis, mediation of the immune response, and the regulation of cell death [1].

Despite their importance in cell metabolism, mitochondria are particularly susceptible to damage, especially that resulting from ROS-induced oxidative stress. Because of this susceptibility to damage, mitochondria are primary targets for environmental toxic chemicals; these typically disrupt the mitochondrial electron transport chain, resulting in reduced ATP production, increased ROS-induced oxidative stress, and increased apoptosis. This in turn suggests a rationale for investigating the potential role of coenzyme Q10 (CoQ10) in mediating such chemical-induced mitochondrial dysfunction.

CoQ10 is usually described as a vitamin-like substance, although by definition, CoQ10 is not a vitamin, since it is synthesised by most tissues within the human body. CoQ10 has a number of functions of vital importance to normal mitochondrial metabolism. Within mitochondria, CoQ10 has a key role as an electron carrier (from complex I and II to complex III) in the mitochondrial electron transport chain during oxidative phosphorylation. It is also involved (as a cofactor of the enzyme dihydroorotate dehydrogenase) in the metabolism of pyrimidines, fatty acids, and mitochondrial uncoupling proteins, as well as in the regulation of the mitochondrial permeability transition pore. CoQ10 serves as an important lipid-soluble antioxidant protecting mitochondrial membranes from ROS-induced oxidative stress [1]. CoQ10 supplementation has been successfully used in both primary and secondary deficiency disorders [2,3].

In the present article, we have, therefore, reviewed the potential role of supplementary CoQ10 in improving mitochondrial function and mediating adverse effects following exposure to a number of environmental toxins, including pesticides, heavy metals, industrial solvents, endocrine-disrupting agents, and carcinogens, as well as the toxic effects of pharmacological drugs and lifestyle toxins.

## 2. CoQ10 and Pesticide/Herbicide Toxicity

Exposure to pesticides can result in a number of adverse effects, including the development of neurological, immune, and hormonal disorders; this is particularly the case in developing countries and for organophosphate-based pesticides [4]. At the cellular level, mitochondria are principal targets for pesticides, many of which inhibit complexes of the electron transport chain (particularly complex l), resulting in reduced ATP generation, increased oxidative stress, and increased apoptosis [5,6]. Mitochondria are particularly susceptible to damage by lipophilic pesticides, which readily access mitochondria via their affinity with the mitochondrial translocator protein binder.

The protective action of supplementary CoQ10 against the adverse effects of pesticide exposure is a well-researched area, at least in animal models (principally rats or mice). In addition to their toxic effects per se in terms of environmental exposure, studies of specific pesticides in these animals serve as preclinical models for various disorders; for example, the administration of rotenone or paraquat in rats has been used as a model system for Parkinson’s disease.

The protective effect of CoQ10 against the adverse effects of a number of pesticides has been investigated, including carbofuran, copper sulphate, diaznon, dicholvos, diquat, mevinphos, paraquat, and rotenone. CoQ10 was usually administered orally or via intraperitoneal (i.p.) injection, either prior to, during, or post-pesticide exposure (see Table 1), with the greatest protective effect generally obtained when CoQ10 was administered prior to pesticide exposure. Beneficial effects following CoQ10 administration typically include reduced oxidative stress, reduced inflammation, improved mitochondrial function, reduced tissue degeneration, and improved tissue function. Individual preclinical studies supplementing CoQ10 are summarised in Table 1.

## 3. CoQ10 and Heavy Metal Toxicity

Exposure to toxic heavy metals typically occurs as a result of contamination from industrial or agricultural activity; arsenic, cadmium, chromium, lead, and mercury are considered to be of particular importance with regard to public health. Heavy metal toxicity depends on a number of factors, including the chemical species; the dose; the route of exposure; the duration of exposure; and the age, gender, genetics, and nutritional status of exposed individuals [19]. Heavy metals can affect the function of tissues throughout the body, as well as the function of various organelles within cells; as noted in the Introduction, mitochondrial function is particularly susceptible to heavy metal-induced oxidative stress [20]. For example, cadmium exerts its toxic action principally by blocking the mitochondrial electron-transfer chain, impairing electron flow through complex III; this in turn results in reduced ATP generation, increased oxidative stress, and apoptosis [21]. Heavy metals preferentially accumulate within the mitochondria, accessed via the calcium transporter (because of their similarity to the Ca^2+^ ion). Preclinical studies demonstrating the protective action of supplementary CoQ10 against the toxic effects of a number of heavy metals are shown in Table 2; typically, this involves reductions in oxidative stress and inflammation, with concurrent improvements in tissue function.

## 4. CoQ10 and Industrial Solvent Toxicity

Industrial solvents include xylene, toluene, benzene, methanol, ethylene glycol, and carbon tetrachloride, all of which are reported to cause mitochondrial dysfunction and increased oxidative stress [31,32,33,34,35,36]. In workers in the paint industry occupationally exposed to xylene, elevated levels of peroxidised lipids in plasma were reduced following CoQ10 supplementation [37]. In a mouse model of benzene-induced immune dysfunction, supplementation with CoQ10 reduced oxidative stress and alleviated damage to spleen and thymus tissues [38]. Using an in vitro mitochondrial respiratory assay, several aromatic industrial solvents, including benzene, were shown to inhibit mitochondrial respiration via their interaction with CoQ10 [39]. In rats, supplementation with CoQ10 reversed methanol-induced retinopathy [40]. Several studies in rats have demonstrated the protective effect of pre-administered or co-administered CoQ10 on carbon tetrachloride toxicity; oxidative stress levels were reduced and liver and heart tissue function improved [41,42,43,44]. It is of note that supplementation with a combination of antioxidants and antioxidant precursors, including CoQ10, vitamin E, selenium, and methionine, improved the clinical status of workers in the gas and oil industries exposed to occupational and environmental stress [45].

## 5. CoQ10 and Aircraft Fume Events

Fume events refer to the contamination of aircraft cabin air by fumes from hydraulic fluid, engine oil, or their thermal degradation products; organophosphates are considered to be the principal contaminants, although substances such as benzene and toluene may also be present [46,47]. The potential role of supplementary CoQ10 in mediating mitochondrial dysfunction and oxidative stress resulting from organophosphate exposure has been reviewed by Mantle and Hargreaves [48].

## 6. CoQ10 and Endocrine Disruptors

Work on CoQ10 and endocrine disruptors has focussed principally on bisphenol A, an industrial chemical used in plastics manufacturing. Bisphenol A is a widely distributed environmental endocrine disruptor linked with reproductive dysfunction. Work in cell culture using the C2C12 cell line (a sub-clone of myoblasts) has shown that bisphenol A inhibits gene expressions related to mitochondrial biogenesis, decreases mitochondrial membrane potential, disrupts lysosomal function, and increases oxidative stress and apoptosis; supplementation with CoQ10 essentially corrected these dysfunctional parameters [49]. Using the nematode *Caenorhabditis elegans*, a model with many genetic and physiological similarities to humans, Hornos-Carneiro et al. [50] reported that supplementation with CoQ10 counteracted bisphenol A-induced reproductive toxicity by reducing mitochondrial dysfunction and oxidative stress, thereby reducing DNA damage. In animal models of endocrine disruptor toxicity, oral pre-administration of CoQ10 (10 mg/kg/day for 14 days) reduced bisphenol A-induced oxidative stress and testicular toxicity in rats [51]. Similar administration of CoQ10 in rats reduced bisphenol A-induced oxidative stress, apoptosis, and testicular damage [52].

## 7. CoQ10 and Carcinogens

Microcystins comprise a group of more than one hundred toxins with carcinogenic action produced by cyanobacteria, of which microcystin-LR is the most commonly occurring. In microcystin-LR-treated mice, co-administration of CoQ10 (10 mg/kg/day, i.m., for 14 days) reduced microcystin-LR-induced toxicity via modulation of the glycolytic–oxidative–nitrosative stress pathway [53]. Mycotoxin ochratoxin, derived from certain types of fungi, is known to induce renal damage and kidney cancer; in rats, co-administration of CoQ10 reduced ochratoxin-induced oxidative stress and renal tissue injury [54].

In rats with mammary carcinoma induced by exposure to 7, 12 dimethyl benz(a)anthracene (DMBA), co-administration of tamoxifen and CoQ10 (40 mg/kg/day for 28 days) reduced oxidative stress and prevented cancer cell proliferation [55]. In rats with azoxymethane-induced colonic premalignant lesions, dietary pre-administration of CoQ10 (200–500 ppm for 4 weeks) suppressed lesion formation, suggesting CoQ10 may be an effective chemopreventive agent against colon carcinogenesis [56].

## 8. CoQ10 and Pharmacological Drug Toxicity

The withdrawal of pharmacological drugs from the market, for example, because of cardiovascular safety concerns, in turn resulting from drug-induced mitochondrial dysfunction, has been highlighted by Varga et al. [57]. The adverse effects of a number of such pharmacological drugs have been addressed by CoQ10 supplementation; these include doxorubicin, paracetamol (acetaminophen), cisplatin, methotrexate, cyclophosphamide, amitriptyline, phenytoin, antibiotics, anaesthetics, and statins.

Evidence for the protective effects of supplementary CoQ10 against cardiotoxicity induced by the chemotherapeutic agent doxorubicin has been obtained mainly from preclinical studies, primarily in rats or mice [58]. Oxidative stress resulting from doxorubicin-induced ROS generation causes disruption of mitochondrial energetics and irreversible damage to mitochondrial DNA; this in turn results in the necrosis of myocytes. CoQ10 prevents damage to heart mitochondria, thereby preventing the development of anthracycline-induced cardiomyopathy.

In rats, administration of CoQ10 (10 mg/kg, i.p.) one hour after exposure to a single dose (700–1200 mg/kg, p.o.) of the analgesic paracetamol (acetaminophen) reduced oxidative stress, apoptosis, inflammation, and liver and kidney tissue damage [59,60]. In mice, supplementation with CoQ10 (5 mg/kg, i.v.) prior to paracetamol administration reduced oxidative stress and hepatic tissue injury [61]; CoQ10 supplementation similarly reduced oxidative stress and hepatic tissue injury when given 1.5 h after paracetamol overdose [62]. CoQ10, given 16 h after overdose, was shown to still be effective at a late stage of paracetamol-induced liver injury, decreasing hepatocyte necrosis and promoting hepatocyte proliferation [63].

Several studies have demonstrated beneficial effects of supplementary CoQ10, alone or in combination, with regard to adverse events associated with the chemotherapeutic agent cisplatin. In rats, co-administration of CoQ10 with cisplatin reduced oxidative stress-induced injury to the retina [64] and ovaries [65]. CoQ10 in combination with epigallocatechin gallate improved mitochondrial function and reduced apoptosis and liver tissue injury in cisplatin-treated rats [66]. Similarly, in cisplatin-treated rats, CoQ10 in combination with multivitamins prevents ototoxicity [67] and in combination with trimetazidine, reduces cisplatin-induced oxidative stress in rat cardiomyocytes [68].

In rats treated with the immunosuppressant methotrexate, co-administration of CoQ10 variously reduced hepatic toxicity via mediation of oxidative stress and inflammation [69], reduced oxidative stress and fibrosis in lung and liver tissue [70], reduced oxidative stress and inflammation in testicular tissue [71], and reduced oxidative stress and inflammation in ovarian and uterine tissues [72]. Similarly, in rats treated with the immunosuppressant cyclophosphamide, co-administration of CoQ10 reduced oxidative stress and neuronal damage in brain tissue [73,74]; reduced oxidative stress and renal tissue damage [75]; and reduced oxidative stress and DNA damage in liver, kidney, and brain tissues [76].

The tricyclic antidepressant amitriptyline impairs mitochondrial function and increases oxidative stress; thus, in patients with depression, mitochondrial mass, ATP, and CoQ10 levels were reduced, and lipid peroxidation levels increased in peripheral blood cells [77]. In cultured human fibroblasts, CoQ10 supplementation improved amitriptyline-induced mitochondrial dysfunction and reduced oxidative stress and apoptotic cell death [78]. In rats, supplementary CoQ10 reduced oxidative stress and prevented cognitive impairment induced by the anti-epileptic drug phenytoin [79].

Because of the evolutionary relationship between mitochondria and bacteria (which share similar DNA and ribosomal structures), mitochondria are particularly sensitive to the adverse effects of most classes of antibiotics [80]. Thus aminoglycosides, macrolides, oxazolidinones, chloramphenicol, clindamycin, tetracyclines, glycylcyclines, fluoroquinolones, rifampicin, bedaquiline, and β-lactams can inhibit mitochondrial translation and other mitochondrial functions due to their interactions with mitochondrial components [81]. The potential for antibiotics to impair mitochondrial function is an important consideration in their clinical use, in turn providing a rationale for the co-administration of CoQ10. For example, in mice, supplementary CoQ10 protected sensory hair cells in the inner ear against neomycin-induced cell death [82]. Similarly, in guinea pigs, CoQ10 administration reduced gentamicin-induced loss of sensory hair cells [83]. In rats, supplementary CoQ10 reduced oxidative stress and improved liver function in hepatotoxicity induced by the antitubercular drug rifamycin [84]. In mouse liver, exposure to the antibiotic chloramphenicol results in oxidative stress-induced morphological and functional changes in mitochondria via the formation of so-called megamitochondria, a process suppressed by pre-treatment with CoQ10 [85].

Another class of pharmacological drugs with the potential to induce adverse effects are anaesthetics. For example, the widely used intravenous anaesthetic propofol may have adverse effects (propofol infusion syndrome) on CNS [86]. In rats, propofol has been shown to interact with CoQ10, thereby impeding the flow of electrons through the mitochondrial respiratory chain and reducing ATP synthesis [87]. In cell culture, the latter adverse effects induced by propofol (particularly on complex I) were negated following supplementation with CoQ10 [88]. In rabbits, co-administration of CoQ10 reduced organ injuries associated with propofol infusion syndrome [89]. Using porcine cardiac mitochondria, the anaesthetics halothane, isoflurane, and sevoflurane have also been shown to inhibit complex I of the mitochondrial respiratory chain [90], and co-administration of CoQ10 has been reported to reverse sevoflurane-induced mitochondrial dysfunction in mice [91].

Statins are drugs used to treat dyslipidemia and reduce the risk of cardiovascular disease; they reduce cholesterol levels by inhibiting the activity of the enzyme HMG-CoA reductase, which also forms part of the pathway involved in CoQ10 biosynthesis. Statin use is associated with a number of adverse effects, the most common of which is statin-induced myopathy. To date, there have been eight randomised controlled clinical trials that specifically investigate the effects of supplementary CoQ10 on statin-induced myopathy; four of these studies have reported decreased muscle pain associated with statin treatment [92,93,94,95], and four studies have reported no reduction in muscle pain [96,97,98,99]. The most recent systematic review concluded that CoQ10 supplementation significantly ameliorates statin-induced musculoskeletal symptoms [100].

There are pharmacological drugs that are known to exert adverse effects by inducing mitochondrial dysfunction and oxidative stress, for which the potential benefit of CoQ10 supplementation has, to date, not been assessed. Examples include gadolinium-based contrast agents used in medical imaging [101], the radiographic contrast agent ioversol [102], and antiretroviral drugs used in HIV therapy [103]. In addition, virtually all psychiatric medications reportedly promote mitochondrial impairment [104,105].

## 9. Lifestyle-Related Toxicants

Lifestyle-related toxicants included in this category include ethanol; nicotine/cigarette smoke; and recreational drugs, such as MDMA, cocaine, and khat. The damaging effects of ethanol on liver function are well known, but it is less well known that ethanol damages every other tissue in the body by inducing mitochondrial dysfunction and oxidative stress [106]. In rats, supplementary CoQ10 reduced ethanol-induced hepatotoxicity via inhibition of the NLRP3/caspase-1/IL-1 pathway [107]. Similarly, in rats, supplementary CoQ10 reduced ethanol-induced neuropathic pain by reducing oxidative stress and inflammation [108]. Treatment of corneal fibroblasts with CoQ10 reduced oxidative stress and apoptosis induced by ethanol; this is of relevance because of the use of ethanol during corneal surgery [109].

Exposure to nicotine or cigarette smoke causes a number of adverse effects within the body, including reductions in bone density and renal tissue injury. Both nicotine and cigarette tar adversely affect mitochondrial function. Using rat brain mitochondria, Cormier et al. [110] showed that nicotine inhibits mitochondrial respiration by binding to complex I of the respiratory chain. Similarly, Pryor et al. [111] demonstrated that cigarette tar inhibits mitochondrial respiration and increases free radical generation in isolated mitochondria. In nicotine-exposed rats, supplementation with CoQ10 improved bone fracture resistance [112]. In cultured rat renal proximal tubule cells, supplementary CoQ10 rescued cells from nicotine-induced oxidative stress and consequent apoptosis [113]. With regard to cigarette smoke, blood CoQ10 levels have been reported to be significantly reduced in smokers [114]. In cigarette smoke-exposed mice, administration of CoQ10 improved mitochondrial function and reduced oxidative stress and apoptosis [115].

With regard to recreational drugs, a number of substances of abuse are known to induce mitochondrial dysfunction, including MDMA [116] and cocaine [117]. In rat brain, administration of CoQ10 attenuated energy dysregulation in the MDMA-induced depletion of brain 5-HT [118]. In mouse brain, supplementary CoQ10 reduced oxidative stress and loss of dopamine induced by cocaine exposure [119]. In mice, administration of CoQ10 reduced hepatic and renal tissue injury induced by the recreational drug khat [120].

Included in this section is the tetrahydropyridine substance MPTP; although MPTP itself is not a recreational drug, it was first identified as a contaminant among drug abusers who had self-administered synthetic heroin. The neurotoxic metabolite of MPTP, MPP+, damages dopaminergic neurons by inducing mitochondrial dysfunction and oxidative stress [121]; this in turn causes a Parkinson’s disease-like disorder, and administration of MPTP in rodents has been widely used as a preclinical model for Parkinson’s disease [122]. A number of preclinical studies have demonstrated the therapeutic effect of supplementary CoQ10 in MPTP models of Parkinson’s disease; for example, in mice, CoQ10 administration reduced MPTP-induced loss of dopaminergic nerve terminals in the striatum [123,124,125].

## 10. Conclusions

The functional characteristics of CoQ10 provide a rationale for its supplementary use in protecting against the adverse effects of xenobiotics [1,3]. Briefly, CoQ10 supports cellular energy production as a crucial electron carrier in the mitochondrial electron transport chain, transferring electrons from complexes I and II to complex III—an essential step in ATP synthesis via oxidative phosphorylation [1]. Both ubiquinone and its reduced form, ubiquinol, are widely interconverted in the body via membrane-bound oxidoreductases and are available in supplement form [1]. The oxidized form, ubiquinone, is used in the electron transport process. The reduced form, ubiquinol, is a potent antioxidant. It directly scavenges reactive oxygen species (ROS); regenerates other antioxidants, such as vitamins E and C [1]; and helps protect cellular and mitochondrial membranes from lipid peroxidation [126,127]. However, when CoQ10 in the mitochondrion becomes excessively reduced, it can contribute to reductive stress, driving the electron transport chain in reverse, ultimately increasing mitochondrial oxidative stress (“reverse electron transport”) [128,129]. Beyond its direct roles in antioxidation and energy production, CoQ10 has many other roles. It helps stabilize mitochondrial and cellular membranes [130,131]. CoQ10 upregulates mitofilin, a key component of the mitochondrial inner membrane organizing system, essential for maintaining cristae architecture and mitochondrial function [132]. It enhances mitochondrial biogenesis and mitochondrial mass by stimulating PGC (Peroxisome proliferator-activated receptor gamma coactivator). PGC-1, particularly PGC-1α, is a master regulator of mitochondrial biogenesis [133,134], driving the transcription of genes involved in mitochondrial replication, energy metabolism, and antioxidant defence. CoQ10 helps maintain the mitochondrial membrane potential (Δψm) [135], which is essential for ATP synthesis, as the electrochemical gradient drives ATP synthase activity. Additionally, CoQ10 inhibits the opening of the mitochondrial permeability transition pore (mPTP) [126,136]—a critical event in cell death pathways. Coenzyme Q10 is an essential component of the lysosomal electron transport chain, facilitating proton translocation and maintaining the proton gradient necessary for lysosomal acidification, which in turn is required to activate lysosomal hydrolases responsible for the breakdown of macromolecular waste into smaller degradation products that can be exported and used in autophagy. Following toxic chemical exposure, there is an increased burden of cellular waste products requiring degradation, which places additional demand on the CoQ10-dependent acidification process; furthermore, some chemicals can directly disrupt lysosomal acidification, further increasing the requirement for CoQ10. When CoQ10 is inadequate, this leads to impaired lysosomal acidification, accumulation of undegraded waste, defective autophagy, cell dysfunction, and cell death [137,138].

In this article, we have reviewed the protective action of supplementary CoQ10 against the toxic effects of a wide variety of xenobiotic substances, including pesticides, heavy metals, industrial solvents, endocrine disruptors, and carcinogens, as well as adverse effects associated with prescription drugs and lifestyle-related toxicants. Most of the studies were carried out in animal models; however, supplementary CoQ10 consistently reduced oxidative stress, apoptosis, and inflammation, while improving mitochondrial function in a number of tissues. The data identified in the present article, therefore, provide a rationale to support further investigation of the potential benefits of supplementary CoQ10 in patients suffering from xenobiotic poisoning. In this regard, it is of note that a randomised controlled clinical trial reported symptomatic benefits following CoQ10 administration in Gulf War veterans, who had been exposed to a variety of toxic chemicals of the types described in this article, resulting in mitochondrial dysfunction [139,140,141]. Finally, although individuals may experience adverse effects following exposure to a wide range of different classes of xenobiotic agents, mitochondrial impairment and oxidative stress underlie the pathogenesis of these adverse effects, irrespective of nominal-specific mechanisms [142].

## Figures and Tables

**Table 1 ijms-26-05788-t001:** Preclinical studies supplementing CoQ10 in pesticide-exposed animal models.

Pesticide	Species	CoQ10 Dose	Outcome	Reference
Carbofuran	Rat	100 mg/kg for 21 days, oral (co-administration)	Liver and kidney tissues protected from oxidative stress and inflammation	Hossain et al. (2023) [7]
Copper sulphate	Rat	10 mg/kg/day, oral, for 7 days (co-administration)	Reduced oxidative stress, reduced inflammation, and reduced cardiotoxicity	Alghibiwi et al. (2025) [8]
Diazinon	Rat	10 mg/kg for 30 days, i.p. (co-administration)	Reduced oxidative stress and neonatal brain damage	Chali et al. (2023) [9]
Dichlorvos	Rat	4.5 mg/kg, i.p., for 12 weeks (pre-administration)	Reduced oxidative stress, reduced neurodegeneration, and improved cognitive function	Binukumar et al. (2012) [10]
Diquat	Mice	20 mg/kg/day, gavage, for 1 week (pre-administration)	Reduced oxidative stress, improved mitochondrial function, and improved renal function	Wu et al. (2024) [11]
Mevinphos	Rat	4 mcg, brain injection (co-administration)	Improved mitochondrial function, improved medullary function, and cardiovascular protection	Yen et al. (2005) [12]
Paraquat	Mice	200 mg/kg for 3 weeks (pre-administration)	Reduced brain protein carbonyl levels and improved behaviour	Attia & Maklad (2018) [13]
Paraquat	Rat	6 mg/kg for 4 weeks, oral (post-administration)	Neurodegeneration halted and motor skills improved	Muthukumaran et al. (2014) [14]
Paraquat	Rat	50 mcg/mL, drinking water (pre-administration)	Oxidative stress reduced and neurodegeneration prevented	Somayajulu-Nitu et al. (2009) [15]
Phosphine (as aluminium phosphide)	Rat	100 mg/kg, i.p. (co-administration)	Reduced oxidative stress, improved mitochondrial function, and improved hepatic function	Hooshangi-Shayesteh et al. (2024) [16]
Rotenone	Rat	100 mg/kg for 7 days (pre-administration)	Reduced oxidative stress and improved brain function	Akinmoladun et al. (2022) [17]
Rotenone	Rat	Dose not stated (pre-administration)	Improved mitochondrial function and reduced dopaminergic neuronal death	Moon et al. (2005) [18]

**Table 2 ijms-26-05788-t002:** Preclinical studies supplementing CoQ10 in animal models of heavy metal toxicity.

Metal Type	Species	CoQ10 Dose	Outcome	Reference
Arsenic (as sodium arsenite; 10 mg/kg/day, oral, for 2 days)	Rat	10 mg/kg/day for 5 days, i.p. (pre-administration)	Reduced oxidative stress, reduced inflammation, and reduced testicular tissue injury	Fouad et al. (2011) [22]
Arsenic (as sodium arsenite; 15 mg/kg for 30 days, oral)	Mouse	200 mg/kg for 30 days, oral (co-administration)	Improved haematological parameters and improved hepatic and renal function	Mwaeni et al. (2021) [23]
Cadmium (0.4 mg/kg, i.p., single dose)	Rat	20 mg/kg, i.m., single dose (pre-administration)	Reduced oxidative stress and reduced haematotoxicity	Paunovic et al. (2017) [24]
Cadmium (25 mg/kg/day, oral, for 15 days)	Rat	10 mg/kg/day for 15 days, oral (co-administration)	Improved semen quality and reduced testicular oxidative stress	Saha et al. (2019) [25]
Cadmium (0.4 mg/kg/day for 3 days, oral)		20 mg/kg/day for 30 days, oral (post-administration)	Reduced oxidative stress and improved semen parameters	Iftikhar et al. (2022) [26]
Cadmium (6.5 mg/kg, i.p., single dose)	Mouse	100 mg/kg day for 14 days, oral (post-administration)	Reduced oxidative stress, reduced inflammation, and reduced cardiotoxicity	Antar et al. (2024) [27]
Lead (as lead acetate, 10 mg/mL/day for 28 days, oral)	Rat	10 mg/kg/day for 28 days, oral (co-administration)	Improved serum lipid profile	Mazandaran et al. (2021) [28]
Mercury (as mercuric chloride, 5 mg/kg for 1 week, oral)	Rat	10 mg/kg for 30 days, oral (post-administration)	Reduced nephrotoxicity	Kadry & Megeed (2022) [29]
Titanium (as titanium dioxide, 50 mg/kg + Cadmium 5 mg/kg for 60 days, oral)	Rat	10 mg/kg for 60 days, oral (co-administration)	Reduced oxidative stress, reduced inflammation, and improved hepatic function	Abd-Elhakim et al. (2023) [30]

## Data Availability

Not applicable.

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
