# Peer review of "Coenzyme Q10 and Xenobiotic Metabolism: An Overview"

_ijms, 2025, doi:10.3390/ijms26125788_

Round 1

Reviewer 1 Report

Comments and Suggestions for Authors

The review presents a cohesive and up-to-date synthesis of the literature on coenzyme Q10 as a countermeasure to xenobiotic toxicity. It successfully integrates data across pesticides, heavy metals, solvents, endocrine disruptors, carcinogens, prescription drugs and lifestyle toxicants into a single mechanistic narrative centred on mitochondrial dysfunction and oxidative stress, giving the field a clear unifying framework. Each xenobiotic class is summarised with species, dose, route, timing and outcome, supported by well-constructed reference tables that allow rapid comparison of study design and endpoints. Citations are current through 2025, and the authors consistently link experimental findings to clinically relevant implications, such as Gulf War illness and statin myopathy, underscoring translational value. The manuscript is logically organised, written in concise scientific prose, and free of ambiguity; key terms are defined, abbreviations are standardised, and conflict of interest is transparently disclosed. These qualities make the paper a reliable reference for toxicologists, pharmacologists and clinicians seeking an authoritative overview of CoQ10’s protective potential.

Author Response

We have gone through the reviewer comments and addressed each.

Reviewer 1: The review presents a cohesive and up-to-date synthesis of the literature on coenzyme Q10 as a countermeasure to xenobiotic toxicity. It successfully integrates data across pesticides, heavy metals, solvents, endocrine disruptors, carcinogens, prescription drugs and lifestyle toxicants into a single mechanistic narrative centred on mitochondrial dysfunction and oxidative stress, giving the field a clear unifying framework. Each xenobiotic class is summarised with species, dose, route, timing and outcome, supported by well-constructed reference tables that allow rapid comparison of study design and endpoints. Citations are current through 2025, and the authors consistently link experimental findings to clinically relevant implications, such as Gulf War illness and statin myopathy, underscoring translational value. The manuscript is logically organised, written in concise scientific prose, and free of ambiguity; key terms are defined, abbreviations are standardised, and conflict of interest is transparently disclosed. These qualities make the paper a reliable reference for toxicologists, pharmacologists and clinicians seeking an authoritative overview of CoQ10’s protective potential

Reviewer 2 Report

Comments and Suggestions for Authors

Overall, this is an interesting and comprehensive study of the effect of Q10 on damage created by toxins.  The only scientific thing that I would have liked to be explored is the hypothesis by which the Q10 does this protection, or at least some speculation that would lead to future experiments to determine the mechanism.  I was thinking that the addition of Q10 allows a more efficient use of the electron transport chain and less superoxide formation by Complex III, but I did not see that directly in the text.  There also may be other ideas around that are more comprehensive and a paragraph describing those ideas would be helpful for the reader.

Other small issues-

In table 2, the "Reduced Oxidataive stress..." in Cadmium is in a different font. 

Line 138- there is an extra letter in the "and" between dysfunction and oxidative stress

Sections 7 and 8 have several 1-sentence paragraphs. These should be consolidated rather than having a 1-sentence paragraph.  The last two sentences in section 7 could be combined to make one paragraph or all of the sentences could just be one big paragraph.

Section 8, the first two sentences can be combined into one paragraph.

Line 190- the I after Q10 is capitalized and needs a comma.  Instead it should be "...rats, Q10, in combination with..."

Author Response

Reviewer 2: Overall, this is an interesting and comprehensive study of the effect of Q10 on damage created by toxins.  The only scientific thing that I would have liked to be explored is the hypothesis by which the Q10 does this protection, or at least some speculation that would lead to future experiments to determine the mechanism.  I was thinking that the addition of Q10 allows a more efficient use of the electron transport chain and less superoxide formation by Complex III, but I did not see that directly in the text.  There also may be other ideas around that are more comprehensive and a paragraph describing those ideas would be helpful for the reader.

We have added a paragraph about the many ways coenzyme Q10 supports cell energy antioxidation, defends against apoptosis, modifies gene expression, enhances mitochondrial biogenesis, protects the mitochondrial membrane potential and mitochondrial permeability transition pore, etc. to provide context for the means by why CoQ10 may protect against adverse xenobiotic effects that operate through such mechanisms. We have placed this paragraph as the first paragraph of the conclusion. We reason that it should come after the evidence that CoQ10 had the stipulated protective effects, at the point the reader may wish to know why. An alternative would be to create a new subsection heading to house either this long paragraph or it could be broken into two (or more) paragraphs.

Other small issues-

In table 2, the "Reduced Oxidataive stress..." in Cadmium is in a different font. Fixed.

Line 138- there is an extra letter in the "and" between dysfunction and oxidative stress. Fixed.

Sections 7 and 8 have several 1-sentence paragraphs. These should be consolidated rather than having a 1-sentence paragraph.  The last two sentences in section 7 could be combined to make one paragraph or all of the sentences could just be one big paragraph. Fixed.

Section 8, the first two sentences can be combined into one paragraph. Fixed.

Line 190- the I after Q10 is capitalized and needs a comma.  Instead it should be "...rats, Q10, in combination with..." Fixed.